# Numerical Simulation of Thermal Therapy for Melanoma in Mice

**DOI:** 10.3390/bioengineering11070694

**Published:** 2024-07-09

**Authors:** Yunfei Zhang, Mai Lu

**Affiliations:** Key Laboratory of Opto-Electronic Technology and Intelligent Control of Ministry of Education, Lanzhou Jiaotong University, Lanzhou 730070, China; 11220828@stu.lzjtu.edu.cn

**Keywords:** magnetic induction hyperthermia, Helmholtz coils, digital mouse models, magnetic nanoparticles

## Abstract

In recent years, the progressively escalating incidence and exceptionally high fatality rate of cutaneous melanoma have drawn the attention of numerous scholars. Magnetic induction hyperthermia, as an efficacious tumor treatment modality, has been promoted and applied in the therapy of some tumors. In this paper, the melanoma atop the mice’s heads was chosen as the research subject, and a magnetic induction hyperthermia approach based on Helmholtz coils as the magnetic field excitation was investigated and designed. The influence of the electromagnetic field and thermal field on organisms was addressed through modeling by COMSOL simulation software. The results showed that the maximum values of induced electric field and magnetic induction strength in mouse tumor tissues were 63.1 V/m and 8.5621 mT, respectively, which reached the threshold value of magnetic field strength required for magnetic induction hyperthermia. The maxima of the induced electric field and magnetic induction intensity in brain tissues are, respectively, 35.828 V/m and 8.57 mT. Approximately 93% of the tumor tissue can reach 42 °C, and the maximum temperature is 44.2 °C. Within this temperature range, a large quantity of tumor cells can be successfully induced to undergo apoptosis without harming normal cells, and the therapeutic effect is favorable.

## 1. Introduction

Cancer is one of the four major non-communicable diseases (NCDs) that cause the most deaths worldwide. According to a report released by The National Cancer Center (NCC) of China in 2024 [1], in China, it was estimated that about 4,824,700 new cancer cases and 2,574,200 new cancer deaths occurred in China in 2022.

Melanoma, a type of basal skin cancer that arises from the cancerous transformation of melanin-converting cells in an organism [2], is mainly affected by the intensity of ultraviolet radiation (UVR) and the duration of exposure of the receptor to UVR [3]. Some statistics show that the survival rate of patients is less than 5% within 5 years, while the average survival period of patients is only about 6 to 9 months [4,5]. Therefore, it can be said that melanoma is the deadliest form of skin cancer. As of 2020, a total of 325,000 new melanoma cases and 57,000 melanoma deaths have been generated globally, and if the incidence of melanoma remains at 2020 levels over the next 20 years, the number of melanoma cases is expected to increase to 510,000 globally, and deaths due to melanoma will reach 96,000 by 2040 [6]. In China, the age-standardized incidence of melanoma is 0.9 cases per 100,000 people, with 20,000 new cases each year [7]. Currently, the mainstream treatment options for melanoma are surgery, immunotherapy, targeted therapy, and hybrid therapy [8].

Common physical therapy treatments for tumors are radiofrequency ablation (RFA) and microwave ablation (MWA) [9], cryoablation based on cryogenic means and emerging technologies of high-intensity focused ultrasound (HIFU), and laser ablation [10]. Magnetic induction hyperthermia as a physical therapy for tumors has received great attention in recent years. The concept of magnetic induction hyperthermia was first introduced by Gilchrist R.K et al. in 1957. They injected magnetic particles into lymph nodes containing metastatic cancer cells, and by heating these injected metal particles by magnetic induction, they observed apoptosis of cancerous cells [11]. Because direct human experimentation is often not possible in biological experiments, mice are widely used as a substitute for humans. In 1996, Kacew and Festing summarized the advantages of using mice in experiments, including the following: metabolic pathways that are similar to those of humans, with many similar anatomical and physiological characteristics; large databases, which are extremely important for the comparative analysis of data; animals can be bred and raised at relatively low cost; and individual species of lab mouse have above-average skin cancer rates, making them suitable as research receptors for skin cancer treatments [12,13]. Experiments on magnetic induction thermotherapy with mice have been widely carried out, and researchers have shown that magnetic induction thermotherapy has significant effects in treating tumors in mice with breast cancer, brain tumors, and liver tumors [14,15,16]. Heat therapy for melanoma has also gained attention from researchers. Tang’s team used large metal heat seeds to heat up skin melanomas, and found that it was possible to cause massive apoptosis of tumor cells at temperatures ranging from 42 °C to 46 °C, while normal tissues were not affected [17].

Magnetic induction hyperthermia has demonstrated unique advantages in tumor treatment, but only thermotherapy with thermoseeds as heat generators has gained widespread popularity in clinical application [18]. However, there is a lack of theoretical research on the clinical application of new materials, such as magnetic fluids. In order to meet the needs of clinical and scientific research, the application of magnetic fluids in thermotherapy will become a development trend. Meanwhile, the application of novel materials will increase the uncertainties in the treatment, for example, whether the temperature of the tumor and its surrounding biological tissues can reach 42 °C without exceeding 46 °C during the treatment process, so as to ensure that apoptosis of the tumor cells is induced without damaging the normal cells, and the degree of uniformity of the heating is also a key point worth studying. In this paper, the numerical conditions of the magnetic induction strength, induced electric field, and thermal field of the organism when using magnetic fluid as a heat source in magnetic induction thermotherapy are investigated, which provide an important reference for the feasibility and safety of the application of magnetic fluid in clinical thermotherapy in the future.

## 2. Materials and Models

### 2.1. Digital Mouse Model

In order to make the model morphologically specific, reference is made here to the anatomically realistic finite element mouse model derived from computed tomography (CT) data by the Biomedical Imaging Group at the University of Southern California [19,20], which visualizes the relative positions of the mouse organs in vivo and provides aid in localizing the mouse organs. Three-dimensional structural diagram of the mouse are shown in Figure 1.

In order to facilitate the computational work of the finite element simulation software, the work of John M. Bernabei et al. [21] is referred to here to optimize the mouse parameters and external dimensions, as shown in Table 1.

Three-dimensional modeling was performed in COMSOL software, and the part of the figure marked in blue is the location of the designed tumor, which is located in the top part of the rat’s head and graded as a moderately malignant tumor [22], with a surface area of 451.52 mm^2^ and a thickness of 2 mm.

Reference [23] gives the dielectric parameters of tissues such as muscles of mice at 100 kHz by actual measurements as the main source of data for this study. Reference [24] gives the dielectric parameters of biological tissues of the human body at 100 kHz by actual measurements, which are used to supplement the dielectric parameters of brain tissues and the skull. The dielectric parameters of the trunk are calculated by combining the dielectric parameters of the lungs, heart, kidneys, muscles, and bones. References [25,26,27] gives the thermal properties of biological tissues of mammals such as mice by practical measurements. The dielectric parameters as well as the thermal properties of the mouse are shown in Table 2 and Table 3 below. The thermal properties of the tumor and the thermophysical parameters of the magneto-fluid were derived from the literature [28,29], shown in Table 4.

### 2.2. Helmholtz Coil Model

A coil is a device that generates a stable magnetic field in magnetic induction thermotherapy, which is related to the quality and efficiency of the treatment and is the most important link in the whole system. Currently, according to the principle of magnetic field generation, there are mainly coil-type and core-type devices for generating magnetic fields in magnetic induction thermotherapy equipment. The magnetic field strength distribution and magnetic line of force distribution generated by the magnetic core-type device are relatively uniform, but the magnetic core-type coil has to have a ferromagnetic material as the magnetic core during the design process, which increases the manufacturing cost, equipment volume, and energy loss to a certain extent, and is not conducive to parameter adjustment. The magnetic field strength distribution and the distribution of magnetic lines of force generated by the coil-type device are relatively less uniform, but it is more convenient to adjust the parameters. Therefore, in this paper, the coil-type Helmholtz coil [30], which has strong parameter adjustability, is used as the generator of the magnetic field.

The Helmholtz coil in COMSOL is shown in Figure 2, where two identical circular coils are placed parallel to each other and the line connecting the center points is kept perpendicular to the parallel plane of the coils. The dimensions of the coils are shown in Table 5.

According to the Biot–Savart law, the strength of the magnetic field generated by a single coil on the center axis is
(1)$$B_{x}=\frac{\mu_{0}NIR^{2}}{2{\left(R^{2}+x^{2}\right)}^{\frac{3}{2}}}$$
where *μ*_0_ is the vacuum permeability; N is the number of turns of the coil; I is the amplitude of the alternating current through the coil in A; R is the radius of the coil in m; and x is the distance from a point on the central axis of the coil to the center point O in m. Then, the magnetic induction at any point on the central axis of the Helmholtz coil is
(2)$$B=B_{R}+B_{L}=\frac{\mu_{0}NIR^{2}}{2{\left(R^{2}+{\left(\frac{R}{2}-x\right)}^{2}\right)}^{\frac{3}{2}}}+\frac{\mu_{0}NIR^{2}}{2{\left(R^{2}+{\left(\frac{R}{2}+x\right)}^{2}\right)}^{\frac{3}{2}}}$$

In order to generate a magnetic field that meets the strength required for thermal therapy, copper is used as the coil winding material and the magnetic field frequency is designed to be 100 kHz; the parameters of the material and the parameters of the energizing current are shown in the Table 6.

## 3. Principles and Methods

### 3.1. Introduction to the Software

COMSOL Multiphysics is based on the finite element method to realize the simulation of real physical phenomena by solving partial differential equations (single field) or systems of partial differential equations (multiple fields). Its basic principle is to discretize the solution domain into a finite number of non-overlapping cells, select some suitable nodes as interpolation points in each cell, rewrite the dependent variable in the partial differential equation to be solved into a system of linear equations composed of interpolation functions based on the values at the nodes, i.e., the so-called stiffness matrix, and finally solve it by appropriate numerical methods, so as to make a continuous infinite-degree-of-freedom problem into a discrete finite-degree-of-freedom problem. The solution steps are as follows:Regional discretization or sub-domain delineation;An interpolating function that approximates the unknown solution in a cell;Establish unit equations and introduce boundary conditions to form a system of equations;Solve the system of equations.

### 3.2. Principles of Magnetic Induction Hyperthermia

Magnetic media in an alternating magnetic field will have losses due to the presence of relaxation, which manifests itself as heat production by absorbing the energy of the magnetic field. The heat generation power of superparamagnetic nanomaterials in an alternating magnetic field can be expressed as follows [31]:(3)$$P_{0}=\mu_{0}\pi H_{0}^{2}\chi_{0}\frac{2\pi f\tau_{eff}}{1+{\left(2\pi f\tau_{eff}\right)}^{2}}$$

Here, μ_0_ is the vacuum permeability, H_0_ is the magnetic field strength in A/m, τ_eff_ is the complex relaxation time, and χ_0_ is the equilibrium magnetization, denoted as
(4)$$\chi_{0}=\chi_{i}\frac{3}{\xi }\left(\coth\xi -\frac{1}{\xi }\right)$$
where ξ = μ_0_M_d_H_0_V / k_d_T is the Langevin parameter; M_d_ is the magnetization intensity of single domain particles in the magnetic fluid, in A/m; V is the volume of a single magnetic nanoparticle, in m^3^; k_d_ is the Boltzmann constant; T is the temperature, in K; and χ_i_ is the initial magnetization rate. Here, according to the medical practice application, take φ = 0.003.
(5)$$\chi_{i}=\frac{\mu_{0}\varphi M_{d}^{2}V}{3k_{b}T}$$

Maxwell’s system of equations is the basis of macroscopic electromagnetic field theory in the following form [32]:(6)$$\begin{array}{l} \nabla \times H=J+\frac{\partial D}{\partial t}\\ \nabla \times E=-\frac{\partial B}{\partial t}\\ \nabla \cdot B=0\\ \nabla \cdot D=\rho \end{array}$$
where H is the magnetic field strength in units of A/m; J is the current density in units of A/m^2^; D is the flux density in units of C/m^2^; E is the magnetic field strength in units of V/m; B is the flux density, i.e., the magnetic induction strength, in units of T; and ρ is the density of the body of charge in units of C/m^3^. In the presence of the medium in question, the above basic equations for the electromagnetic field are incomplete, and both E and B are related to the properties of the medium. Therefore, it is also necessary to add the properties of the medium in the equation; for an isotropic medium, these are
(7)$$\begin{array}{l} D=\varepsilon E\\ B=\mu H\\ J=\sigma E \end{array}$$
where ε is the dielectric constant of the medium in F/m; μ is the magnetic permeability of the medium in H/m; σ is the electrical conductivity of the medium in S/m. With the above equation, we can explain the principle of heat generation of magnetic nanofluids injected into biological tissues. And in order to further describe the specific effect of nanomaterial heat generation in living organisms, Pennes’ bioheat equation is a commonly used model to study the heat transfer in biological tissues, which is expressed as follows [33,34]:(8)$$\rho c\frac{\partial T}{\partial t}=\nabla \cdot \left(k\nabla T\right)+\rho_{b}C_{b}\omega_{b}\left(T-T_{b}\right)+Q_{m}+\alpha P_{0}$$
where ρ is the density of the biological tissue in kg/m^3^; c is the specific heat capacity of the biological tissue in J/(kg∙K); k is the thermal conductivity of the biological tissue in W/(m∙K); ρ_b_ and C_b_ are the blood density and specific heat capacity of the biological tissue in kg/m^3^, J/(kg∙K), respectively; ω_b_ is the blood perfusion rate; T_b_ is the temperature of the blood in the biological tissue in K; Q_m_ is the heat of the metabolism of the organism in K; α is the correction factor with a value of 0.55; and P_0_ indicates the effect of the external heat source, where it is the relaxation loss, that is, the power of nanoparticle heat generation.

### 3.3. Principle Verification

In order to verify the validity of the research method in this paper, the study of Lin [35] et al. is used here for validation, and the modeling is carried out according to the data in the literature. The results show that under the physical conditions set in the literature, the temperature in the center of the tumor reaches more than 42 °C. The results of the theoretical calculations were the same as the literature results, as shown in Figure 3, which means that the method used in this study is feasible.

## 4. Results and Analysis

### 4.1. Electromagnetic Field Distribution in Space

In order to ensure that the spatial magnetic field generated by the coil can meet the needs of thermal therapy, the spatial magnetic field generated by the Helmholtz coil is first studied according to the established parameter settings.

From Figure 4a,b, it can be seen that the Helmholtz coil can generate an alternating magnetic field of uniform magnitude of magnetic induction near the geometric center point of the coil, with a magnitude of 8.5621 mT. According to the literature [36,37], the typical value of the magnetic field strength required for magnetic induction hyperthermia is about 10 mT. Taking into account the depth of the treatment in the simulations in this paper as well as the size of the model, it can be assumed that the Helmholtz coil generates a magnetic field that satisfies the requirements of magnetic induction thermotherapy.

In addition, in space, the variation in the size of the magnetic induction along the x-axis direction is small in the range of −50 mm to 50 mm, which means that a tumor of the size set in this paper placed in a spatial magnetic field is subject to a relatively small amount of influence on the heating intensity of the tumor due to the non-uniform distribution of the magnetic field. From Figure 4c,d, it can be seen that the induced electric field intensity is maximum at the coil and minimum at the coil circular gap. It is represented in the YZ plane as the induced electric field strength starts from the coil and gradually decreases in a circular shape.

### 4.2. Spatial Electromagnetic Field Distribution after Adding the Mouse Model

#### 4.2.1. Magnetic Field Density Distribution

The geometric center of the skin melanoma on the top of the rat’s head was placed at the geometric center point of the Helmholtz coil, the spatial origin, and added to the spatial magnetic field.

From Figure 5a,b, it can be seen that the effect on the spatial distribution of the magnetic field is small after the addition of the mouse to the model. From Figure 5d, it can be seen that the magnetic induction intensity at the center point of the melanoma is 8.5629 mT, which is slightly larger than that before the addition of the mouse to the model. However, the sparseness of the spatial distribution of the magnetic induction intensity did not change. This is because the magnetic permeability of biological tissues is very close to that of air, and the effect on the magnetic field is almost negligible. In the numerical calculations, the default magnitude of both is one. Therefore, after adding the mouse model, the magnetic field generated by this coil still meets the needs of magnetic induction thermotherapy, reaching the typical values of magnetic field strength mentioned in the literature [36,37], proving the effectiveness of the coil design. The magnetic field strength at the center point of the melanoma was calculated as 6814.2 A/m.

#### 4.2.2. Induced Electric Field Strength Distribution

In this study, the relative permittivity of air is set to be one. In contrast, the relative permittivity of biological tissues is significantly greater than one. The disparity between the two is substantial; hence, the induced electric field produced by the electromagnetic field in space undergoes a considerable distortion when passing through biological tissues. This is depicted in Figure 6a,b, where the induced electric field, which was initially uniformly distributed along the central axis of the Helmholtz coil, varies along the positional layout of the mouse when the mouse model is incorporated. The intensity of the induced electric field on the surface of the mouse’s body is considerably larger than that within the body. From the induced E-field strength of the melanoma in Figure 6d, the maximum value of the induced electric field strength in melanoma was 63.1 V/m, the minimum value of the induced electric field strength was 33.5 V/m, and the induced electric field strength at the center point of the tumor was 55.994 V/m. These results can offer valuable references for experimental design and safety analysis.

#### 4.2.3. Temperature Field Simulation

According to the literature [16,38], the duration of magnetic induction thermotherapy in medical experiments is usually 240 s to 600 s. In this study, 300 s was selected as the heating time in the simulation, and 37 °C was set as the starting temperature of the simulation.

Here, it is necessary to assume that the magnetic nanomaterials are uniformly injected in the melanoma. From the previous section, the calculated magnetic field strength at the center point of the melanoma is 6814.2 A/m, which is brought into Equation (3) to obtain the heating power of the ferromagnetic nanomaterials of 673,920 W/m^3^, which is used to set the heat source, and the bioheat transfer module in the software is used for the calculation.

As shown in Figure 7a,b, the center temperature of melanoma was 43.9 °C, which could meet the demand of thermotherapy. As can be seen in Figure 7c, the percentage of melanoma tissues with temperatures higher than 42 °C (thermotherapy temperature) is about 93%. According to the literature [36], due to the different tolerance of different cells to high temperatures, cancer cells will be induced to undergo apoptosis by high temperature, while normal cells will not be affected at this temperature. It indicates that melanoma tissues can basically reach the therapeutic temperature and be guided to apoptosis, obtaining an excellent therapeutic effect. Thus, it proves that the experiment designed in this paper can realize the targeted therapy for cancerous tissues.

As shown in Figure 8, the temperature of the tumor site rises rapidly to 42 °C in 0–100 s. At this time, the heat exchange between the melanoma tissue injected with magnetic nanomaterials and the outside world is low. After 100 s, the rate of warming at the tumor site gradually decreased due to the accelerated heat exchange between the melanoma tissue and other biological tissues, which had an effect on the distribution of the temperature field at the melanoma site, and finally, the temperature at the center point of the tumor reached 43.9 °C and the temperature at the top of the tumor reached 44.2 °C, which verified that the melanoma could be effectively heated. The temperature of the skin tissue at the bottom part of the melanoma also increased with the heat conduction process of the organism, and rose to 41 °C after 300 s. At this temperature, the different effects of temperature on normal and tumor tissues can be used to induce apoptosis in tumor tissues. According to the literature [39,40], in addition to the property of directly inducing apoptosis in cancer cells, high temperatures can modulate the activity of various types of immune cells, including antigen-presenting cells and T-cells, and activate the anti-tumor immune system. In addition, high temperature can induce the survival of cancer cells to form more mature cell types, thus inhibiting the self-renewal of cancer cells. Taken together, these results demonstrate that magnetic induction thermotherapy is effective when applied to the experiments designed in this paper.

#### 4.2.4. Distribution of Physical Fields within the Brain Tissue of an Organism

Due to the relative proximity of the melanoma to the brain tissue of the organism, it is necessary to study the distribution of electromagnetic fields in the brain tissue of mice in order to avoid additional bioelectromagnetic effects on the organism caused by the electromagnetic fields used in the performance of thermotherapy.

Figure 9 shows the distribution of the electromagnetic field in the brain tissue and skull of the mouse, and it can be seen that the magnetic induction intensity in the brain of the mouse is between 8.54 mT and 8.57 mT; because the skull has the ability of electrostatic shielding, the maximum value of the induced electric field intensity of the brain tissue, 35.828 V/m, is located close to the top of the head melanoma, and the minimum value of 1.49 V/m is located away from the top of the head melanoma, which proves that the injection location of magnetic nanoparticles has a significant effect on the electric field distribution in the tissues of the organism. The induced electric field strength at the center point of the brain tissue was 10.496 V/m.

Figure 10 shows the distribution of the temperature field in the brain tissue and skull of the mouse, and it can be seen from the results that the highest temperature in the brain tissue is located in the direction close to the tumor within the brain tissue, with a maximum temperature of 39.8 °C and a minimum temperature of 37 °C. The portion of brain tissue with temperature above 38.5 °C is about 10% of the overall brain tissue, which proves that the effect of thermotherapy on the temperature of brain tissue is small and the safety is guaranteed.

## 5. Discussion

Magnetic induction thermotherapy has received more attention in recent years because of its excellent therapeutic effect and economy, and it can be used as a widespread universal tumor treatment. Moreover, with the development of magnetic nanomaterials, magnetic induction thermotherapy technology has been further advanced. This paper introduces the application of magnetic induction thermotherapy technology in tumor treatment and establishes the relationship between the three physical fields of magnetism–electricity–heat through mathematical formulas. The rat model and coil model were established based on real data, and the alternating current parameters to generate the electromagnetic field required for magnetic induction thermotherapy were designed. After adding the rat tissue dielectric parameters and thermal characteristic parameters, the electromagnetic field and temperature field distributions in the rat were calculated by COMSOL physical simulation software (version 6.1).

The results show that the Helmholtz coil can generate a magnetic field of uniform size and direction near the center point with a magnetic induction strength of 8.5621 mT. After adding the rat model, the magnetic induction strength near the center point becomes 8.5629 mT, and the distribution of the magnetic lines of force in the space is changed because of the difference in dielectric properties of the rat tissues and the air, which is expressed as a change in the trend of the rat’s torso. Meanwhile, the maximum induced electric field strength of the magnetic field in the tumor was 63.1 V/m, and the minimum induced electric field strength was 33.5 V/m. The maximum induced electric field strength of the rat brain was 35.828 V/m, located near the top of the head melanoma, and the minimum value was 1.49 V/m, located far away from the top of the head melanoma due to the electrostatic shielding ability of the skull, and the location of the magnetic nanoparticles’ injection had a significant effect on the electric field distribution in the organism. The electric field distribution of the magnetic nanoparticles had a significant effect. After 100 s of electromagnetic field action, the temperature of the tumor center can rise to 42 °C, and at 300 s, the temperature of the melanoma center point rises to 43.9 °C, and the temperature of the apex rises to 44.2 °C, and at the same time, about 93% of the tumor tissues can reach 42 °C, which meets the conditions of tumor apoptosis.

This study utilized a specific Helmholtz coil with 180 turns and current excitation at 8A and 100 kHz exclusively for the model presented in this paper. Should a different tumor model be encountered, it will be necessary to adjust the dimensional parameters and turns of the coil as well as the current excitation. The objective is to achieve high temperatures in the mixed zone of tumor tissue and normal tissue during the heating process, thereby leveraging temperature variability to induce apoptosis in cancer cells.

As a mature technology, magnetic induction thermotherapy not only provides an important solution for clinical tumor treatment but also makes non-invasive tumor treatment possible. The heating method of the tumor and the injection method of magnetic nanoparticles will affect the therapeutic effect. Moreover, when the tumor tissue is heated, the heat conduction will cause the surrounding biological tissues to warm up rapidly, which may cause thermal fatigue or even damage to the normal biological tissues when the heating is not proper. In order to promote the practical application of this technology on a wide scale, safety guidelines and sufficient theoretical research are indispensable, and the simulation results in this paper can provide an important reference.

## Figures and Tables

**Figure 1 bioengineering-11-00694-f001:**
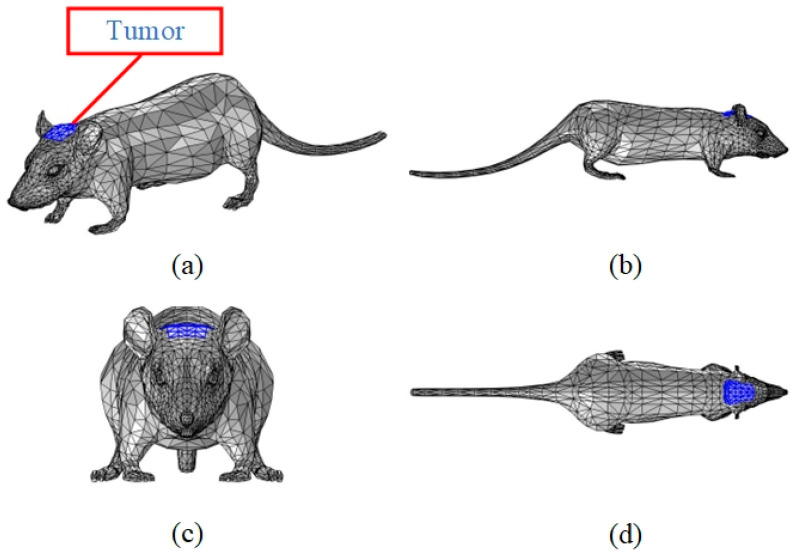
Three-dimensional structural diagram of the mouse. (**a**) Main view; (**b**) XZ-plane view; (**c**) YZ-plane view; (**d**) XY-plane view.

**Figure 2 bioengineering-11-00694-f002:**
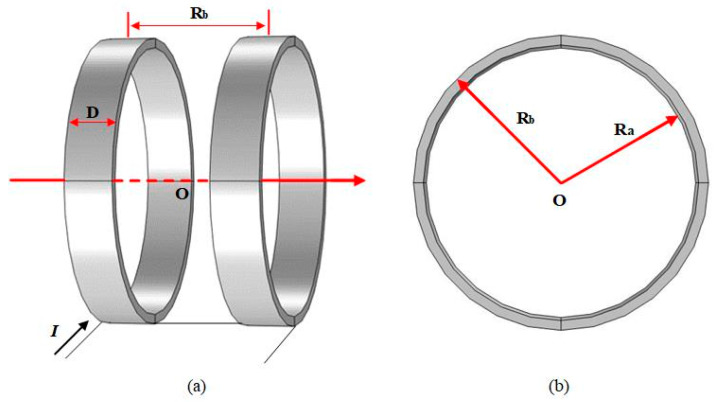
Helmholtz induction coil. (**a**) Main view; (**b**) YZ-plane view.

**Figure 3 bioengineering-11-00694-f003:**
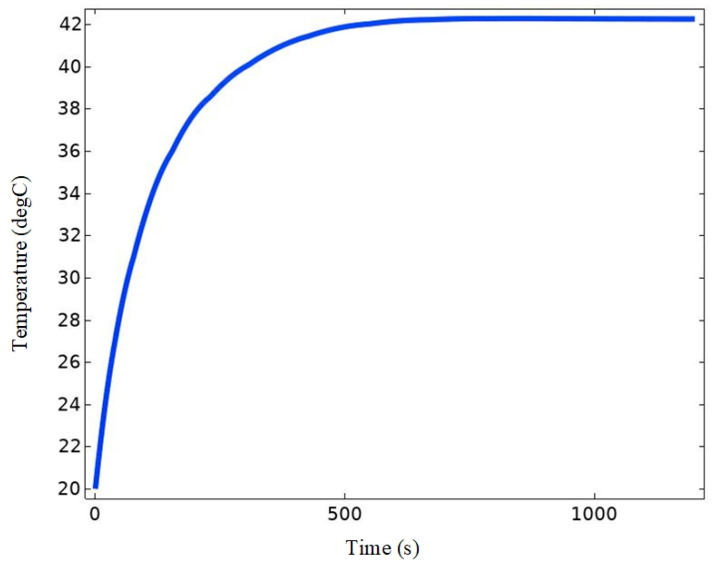
Theoretical calculation of the temperature change at the center point of a tumor.

**Figure 4 bioengineering-11-00694-f004:**
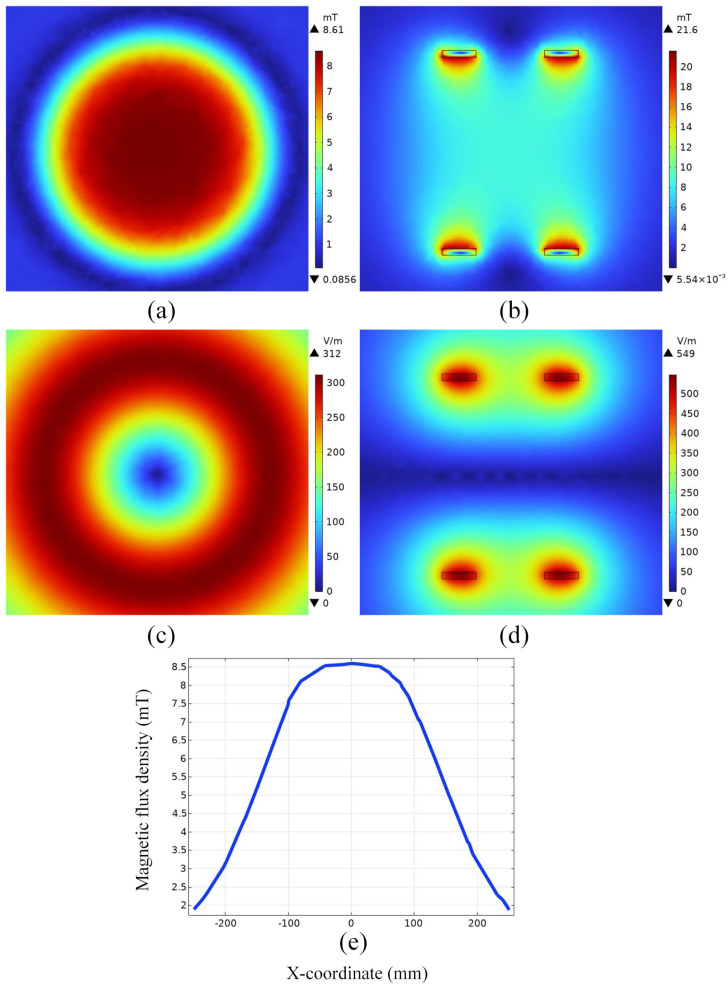
Distribution of electromagnetic fields in space. (**a**) Magnetic induction intensity distribution in the YZ section; (**b**) magnetic induction intensity distribution in the XZ section; (**c**) distribution of induced electric field strength in the YZ section; (**d**) distribution of induced electric field strength in the XZ section; (**e**) distribution of magnetic induction along the x-axis.

**Figure 5 bioengineering-11-00694-f005:**
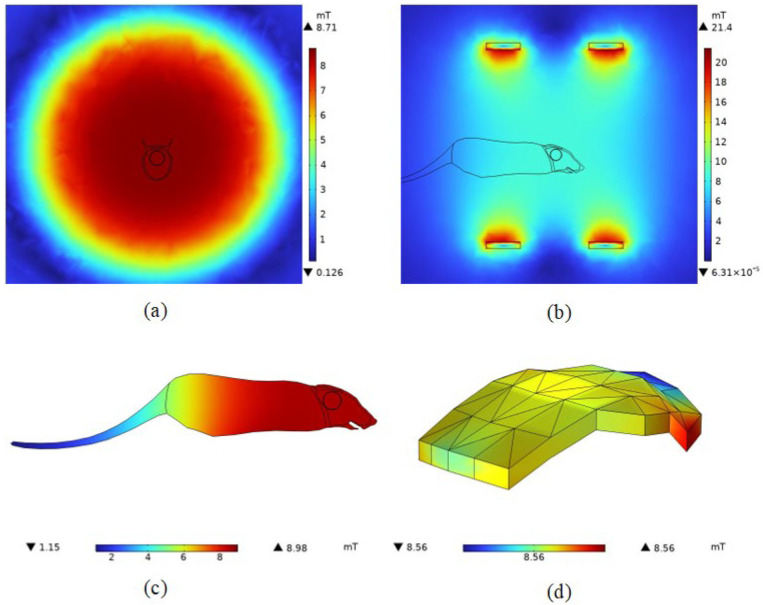
Distribution of magnetic induction intensity in mouse in Helmholtz coils. (**a**) Magnetic induction intensity distribution in the YZ section; (**b**) magnetic induction intensity distribution in the XZ section; (**c**) magnetic induction intensity distribution in mouse; (**d**) magnetic induction intensity body distribution in tumors.

**Figure 6 bioengineering-11-00694-f006:**
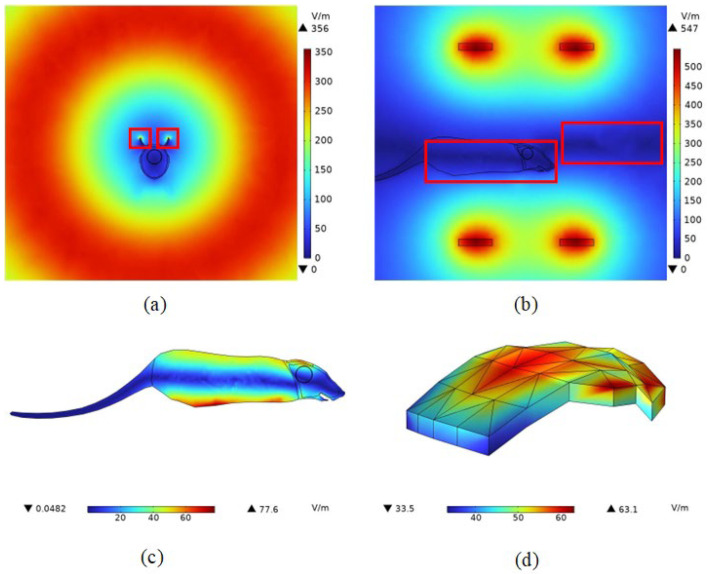
Distribution of induced electric field strength in mouse in a Helmholtz coil. (**a**) Magnetic induction intensity distribution in the YZ section; (**b**) magnetic induction intensity distribution in the XZ section; (**c**) magnetic induction intensity distribution in mouse; (**d**) magnetic induction intensity body distribution in tumors.

**Figure 7 bioengineering-11-00694-f007:**
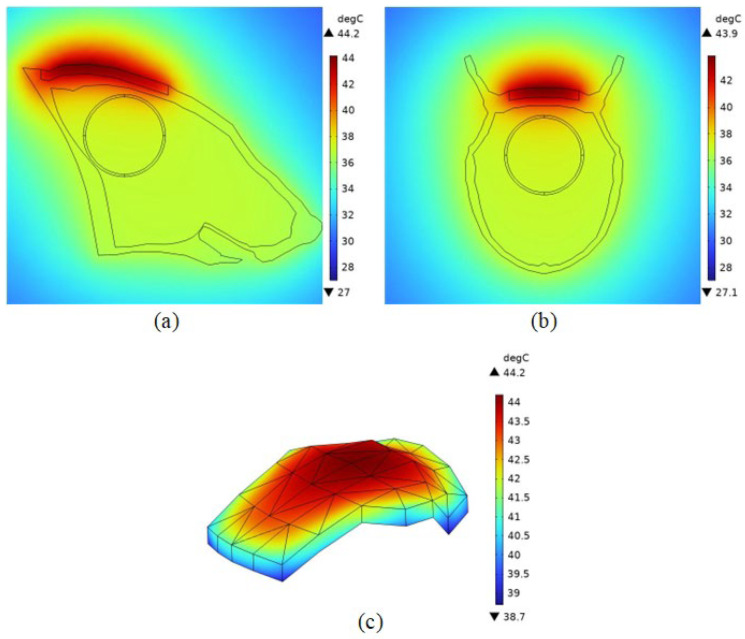
Temperature field distribution in mouse after heating. (**a**) Temperature distribution in the YZ section; (**b**) temperature distribution in the XZ section; (**c**) temperature distribution in tumors.

**Figure 8 bioengineering-11-00694-f008:**
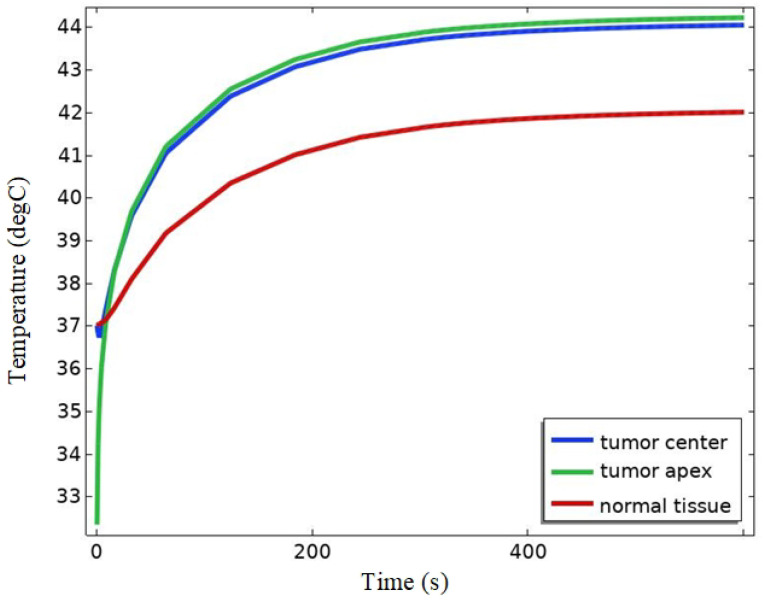
Temperature changes in melanoma and normal biological tissues.

**Figure 9 bioengineering-11-00694-f009:**
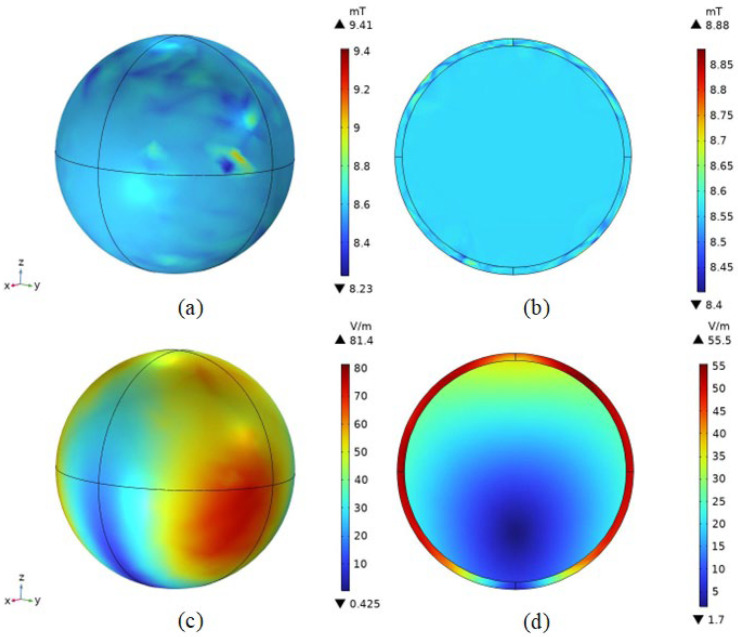
Electromagnetic field distribution in brain tissue and intracranial bone. (**a**) Induced electric field strength distribution; (**b**) induced electric field strength distribution in the YZ section; (**c**) induced electric field strength distribution; (**d**) induced electric field strength distribution in the YZ section.

**Figure 10 bioengineering-11-00694-f010:**
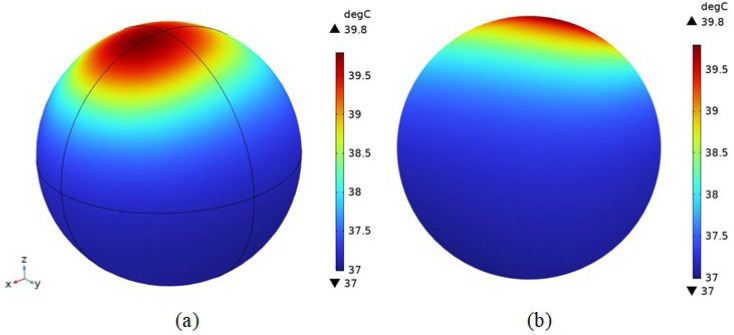
Distribution of temperature field in brain tissue and intracranial bone. (**a**) Temperature field distribution; (**b**) temperature field distribution in the YZ section.

**Table 1 bioengineering-11-00694-t001:** Parameters of external dimensions of mouse.

Dimension	Numeric (mm)
length (including tail)	335
width	67
height	69

**Table 2 bioengineering-11-00694-t002:** Dielectric parameters of biological tissues.

Tissue	Conductivity (σ/(S·m^−1^))	Relative Permittivity
scalp	0.000451	1120
torso	0.304169	5411.25
limbs	0.27945	3670
skull	0.0208	228
brain tissue	0.1079	2665
muscle	0.362	8090

**Table 3 bioengineering-11-00694-t003:** Thermophysical parameters of biological tissues.

Tissue	Density (kg·m^−1^)	Specific Heat Capacity (J·kg^−1^·K^−1^)	Thermal Conductivity (W·m^−1^·K^−1^)	Perfusion (s^−1^)	Metabolic Fever (W·m^−1^)
scalp	1109	3391	0.37	0.02	1620
muscle	1090	3421	0.49	0.00869	480
skull	1908	1313	0.32	0.000436	610
brain tissue	1043	3639.5	0.515	0.00883	7100

**Table 4 bioengineering-11-00694-t004:** Thermophysical parameters of tumor tissues and magnetic nanofluids.

Tissue	Density (kg·m^−1^)	Specific Heat Capacity (J·kg^−1^·K^−1^)	Thermal Conductivity (W·m^−1^·K^−1^)	Perfusion (s^−1^)	Metabolic Fever (W·m^−1^)
magnetic nanofluids	5180	4000	40	-	-
tumor tissue	1060	3650	0.535	0.01392	5790
mixed tissue	1072.4	3651.1	0.5316	-	-
magnetic nanofluids	5180	4000	40	-	-

**Table 5 bioengineering-11-00694-t005:** Coil parameters.

Turns of Each Coil	Width (D)	Outer Diameter (R_a_)	Inner Diameter (R_b_)
180	0.05 m	0.15 m	0.14 m

**Table 6 bioengineering-11-00694-t006:** Current parameters and coil parameters.

Frequency	Amplitude of Current	Relative Permeability	Relative Permittivity	Conductivity
100 kHz	8 A	1	1	5.998e7 S/m

## Data Availability

All data are available from the corresponding author upon request.

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
