# Peer review of "Numerical Simulation of Thermal Therapy for Melanoma in Mice"

_bioengineering, 2024, doi:10.3390/bioengineering11070694_

Round 1
Reviewer 1 Report
Comments and Suggestions for Authors
1. Abstract: The abstract provides specific values (e.g., 63.1 V/m, 8.5621 mT, 42 ℃), but it doesn't explain what these values imply in the context of treatment efficacy and safety. There's a need for a brief explanation of why these values are significant.
2. Introduction: There is some redundancy in the information presented, such as multiple mentions of melanoma's lethality and its potential to metastasize. Streamlining the content would improve the flow.
3. Introduction: Some sentences are ambiguous or unclear, such as "the application of novel materials will increase the uncertainty factor in treatment," which could be clarified for better understanding.
4. Method: Can you provide more detail on the setup and configuration of the Helmholtz coil model?
5. Method: Can you explain how the heating time of 300 s and other simulation parameters were chosen? What is the relevance of these parameters to clinical or experimental settings?
6. Results: Can you provide a more detailed explanation of the changes in magnetic induction and electric field strength distribution after adding the mouse model? Why are these changes significant?
7. Results: Can you provide more context on the implications of the temperature distribution results, particularly how the temperature affects tumor and normal tissues differently?
8. Results: Can you clarify the implications of the temperature distribution results? How do these results support the effectiveness of the proposed magnetic induction hyperthermia technique?
Comments on the Quality of English LanguageMinor editing of English language required.
Reviewer 2 Report
Comments and Suggestions for Authors
The article under the title “Numerical Simulation of Thermal Therapy for Melanoma in Mouse” by Zhang and Lu presents interesting results on simulation of thermal therapy for melanoma. The authors have included up-to-date references and explained the results well. This article is of potential interest to the readers of Bioengineering. There are several points that should be addressed by the authors before final consideration for publication. My recommendation is MAJOR REVISION.
The following points should be addressed by the authors:
1. The authors should check the second sentence of the introduction, as 2019 passed five years ago
2. The authors should specify how were the numerical parameters of biological tissues selected.
3. The authors should explain if Figure 3 represents the experimental or theoretical data.
4. The authors should discuss if the validity of the proposed model changes significantly with the preselected size and if, based on size, the properties of the coil can be determined.
5. What are some of the drawbacks of using thermal therapy?
Round 2
Reviewer 2 Report
Comments and Suggestions for Authors
The authors have answered all of the questions. The article is suitable for publication.